# Lipid Nanoparticles Functionalized with Antibodies for Anticancer Drug Therapy

**DOI:** 10.3390/pharmaceutics15010216

**Published:** 2023-01-08

**Authors:** Ana Camila Marques, Paulo C. Costa, Sérgia Velho, Maria Helena Amaral

**Affiliations:** 1UCIBIO—Applied Molecular Biosciences Unit, MEDTECH, Laboratory of Pharmaceutical Technology, Department of Drug Sciences, Faculty of Pharmacy, University of Porto, R. Jorge Viterbo Ferreira 228, 4050-313 Porto, Portugal; 2Associate Laboratory i4HB—Institute for Health and Bioeconomy, Faculty of Pharmacy, University of Porto, R. Jorge Viterbo Ferreira 228, 4050-313 Porto, Portugal; 3i3S—Instituto de Investigação e Inovação em Saúde, University of Porto, R. Alfredo Allen 208, 4200-135 Porto, Portugal; 4IPATIMUP—Institute of Molecular Pathology and Immunology of the University of Porto, R. Júlio Amaral de Carvalho 45, 4200-135 Porto, Portugal

**Keywords:** cancer, active targeting, functionalization, antibodies, lipid nanoparticles, SLN, NLC, liposomes, antibody-conjugated nanoparticles

## Abstract

Nanotechnology takes the lead in providing new therapeutic options for cancer patients. In the last decades, lipid-based nanoparticles—solid lipid nanoparticles (SLNs), nanostructured lipid carriers (NLCs), liposomes, and lipid–polymer hybrid nanoparticles—have received particular interest in anticancer drug delivery to solid tumors. To improve selectivity for target cells and, thus, therapeutic efficacy, lipid nanoparticles have been functionalized with antibodies that bind to receptors overexpressed in angiogenic endothelial cells or cancer cells. Most papers dealing with the preclinical results of antibody-conjugated nanoparticles claim low systemic toxicity and effective tumor inhibition, which have not been successfully translated into clinical use yet. This review aims to summarize the current “state-of-the-art” in anticancer drug delivery using antibody-functionalized lipid-based nanoparticles. It includes an update on promising candidates that entered clinical trials and some explanations for low translation success.

## 1. Introduction

Chemotherapy, whether alone or combined with other therapies, is the norm for cancer treatment. However, the widespread push to develop nanotechnology-based drug delivery systems comes from the need to overcome the limitations of systemically administered chemotherapeutics, namely short blood circulation time, non-specific distribution in the body, and the development of drug resistance. The size, shape, and surface of nanoparticles can be tailored to escape immediate clearance and enable an efficient delivery to the tumor site [1].

In contrast to normal tissues and organs, many solid tumors have leaky vasculature, along with defective lymphatic drainage, which facilitates the passive accumulation of nanoparticles (100–400 nm in diameter) in the tumor interstitium by the enhanced permeability and retention (EPR) effect [2].

Although passive targeting still occurs initially, active targeting is envisioned as the most promising strategy for improving binding affinity and specificity for tumor cells. For that, the nanoparticle surface can be modified with ligands that bind to receptors overexpressed in angiogenic endothelial cells or cancer cells [3]. In the case of targeting tumor vasculature or certain hematological malignancies, achieving sufficient binding affinity for endothelial cells is critical due to the hemodynamics that nanoparticles experience. Typically, active targeting involves the conjugation of nanoparticles to one or more targeting moieties that interact specifically with receptors that are either uniquely expressed or overexpressed in the tumor compared to normal tissues. If internalizing receptors are targeted, the ligand facilitates the transport of nanoparticles into the cells through a specific pathway once they reach the tumor [4]. For intracellular drug delivery, nanoparticles should be internalized quickly via receptor-mediated endocytosis after specific ligand–receptor interaction. The ability of such actively targeted nanoparticles to bypass drug efflux pumps alleviates the emergence of multidrug resistance [5]. Instead, nanoparticles targeting non-internalizing receptors will remain attached to the target cell surface and release the drug outside, which may also kill nearby cancer cells by the “bystander effect” [6].

In this review, we provide a brief overview of nanoparticle functionalization with antibodies, focusing on the use of these targeting ligands and the most common coupling strategies. Then the current “state-of-the-art” in anticancer drug delivery using antibody-functionalized lipid-based nanoparticles is summarized, including an update on candidates that have entered the clinical testing phase.

## 2. Functionalizing Nanoparticles with Antibodies

The repertoire of ligands conjugated to tumor-targeted nanoparticles is greatly expanded (e.g., antibodies, aptamers, peptides, polysaccharides, and small molecules, such as folate). Among them, antibodies, also known as immunoglobulins (Ig), have gained considerable popularity because of their unique in vivo properties and high specificity [7].

Within the five classes of immunoglobulins (IgG, IgA, IgM, IgD, and IgE) distinguished by the type of heavy chain, IgG is the most abundant in human serum [8]. The IgG molecule is a heterodimeric protein composed of two light (L) chains and two heavy (H) chains. Whereas the light chain consists of one *N*-terminal variable (V_L_) domain and one *C*-terminal constant (C_L_) domain, each heavy chain contains one variable (V_H_) and three constant (C_H_1, C_H_2, and C_H_3) domains. The IgG Y-shaped structure can be divided into two fragments connected by a very flexible hinge region: the antigen-binding fragment (Fab) region, corresponding to the two arms of the antibody molecule, and the fragment crystallizable (Fc) region, referring to its stem region. The *N*-terminal ends of the light and heavy chains collectively form the antigen-binding site, where a total of six hypervariable amino acid sequences termed “complementary determining regions” reside [9,10]. Several functional groups on the amino acids of antibodies can participate in conjugation, namely amino (*N*-terminal or lysine side chain), sulfhydryl (cysteines in the hinge region), and carboxyl (*C*-terminal or aspartic and glutamic acids side chain). Additionally, carbohydrate residues in the C_H_2 domains can be reactive after the periodate oxidation of *cis*-diols to aldehydes [11].

Considering the hydrodynamic radius of the antibody (~20 nm), the size of the functionalized nanoparticle is expected to increase up to 40 nm in proportion to the number of ligands [12]. Since smaller particles allow for deeper penetration into the tumor, antibody fragments (e.g., antigen-binding fragments and single-chain variable fragments) offer a clear advantage over the whole antibody. The primary way to produce an antigen-binding fragment (Fab) is antibody cleavage using proteases, such as papain for monovalent Fab fragments or pepsin for F(ab)_2_ fragments, in which the two arms remain linked. The F(ab)_2_ dimer may then be reduced to yield two Fab’ fragments with *C*-terminal sulfhydryl groups [13]. Alternatively, the V_H_ and V_L_ domains coupled by a flexible and short peptide linker make up the single-chain variable fragment (scFv), commonly obtained by phage display or cloning from mouse hybridoma [14].

To improve physicochemical properties and tumor-targeting accuracy, surface conjugation of functional groups or biomolecules to nanoparticles—also called functionalization—is being extensively studied.

The chemistry behind functionalization with antibodies includes adsorption, covalent conjugation (carbodiimide, maleimide, and “click” chemistries), and avidin–biotin interaction and was reviewed in detail by Marques et al. [15]. It is established that the coupling method should ensure a stable bond and allow for control of the ligand density. Ideally, the Fc region would be attached to the nanoparticle surface, leaving the Fab region oriented in such a way that interaction with the antigen is possible [16]. Although adsorption (i.e., physical adsorption and ionic binding) is the simplest and less time-consuming technique, covalent bonds outperform hydrophobic and electrostatic interactions in terms of stability and reproducibility. As such, carbodiimide and maleimide chemistries persist across the literature, as they benefit from easy-to-follow protocols and acceptable conjugation efficiency at a relatively low cost. In the case of nanoparticles containing surface carboxyl groups, these groups are activated by using 1-ethyl-3-(-3-dimethylaminopropyl) carbodiimide (EDC) and *N*-hydroxysuccinimide (NHS) before reacting with the primary amines of antibodies. However, other coupling strategies that result in oriented immobilization are preferred. More precisely, site-specific free sulfhydryl groups that are generated by antibody reduction or thiolation in the Fc region can be conjugated to the maleimide-activated amino groups of the nanoparticle, yielding a thioether linkage. Another option is based on the non-covalent, albeit very strong, interaction between strept(avidin)-coated nanoparticles and Fc-specific biotinylated antibodies. Even so, compared to direct covalent coupling, the final avidin–biotin complex requires an expensive multistep protocol with less efficient antibody binding, as observed by Wartlick et al. [17].

## 3. Antibody-Functionalized Lipid Nanoparticles for Anticancer Drug Delivery

Particles within the size range of 10 to 1000 nm are defined as nanoparticles. Therapeutic agents (i.e., drugs, proteins, and genetic material) can also be adsorbed or chemically conjugated to the nanoparticle surface, but those that are dissolved, entrapped, or encapsulated into the nanoparticle will benefit from enhanced chemical stability and protection against degradation [18]. Another distinctive feature of nanoparticles is the large surface area/volume ratio compared to bulk materials upon reduction of particle size to the nanoscale [19]. In view of a highly customizable surface, nanoparticles can be modified to avoid major obstacles to successful delivery. For instance, as surface coating with polyethylene glycol (PEG) imparts “stealth” properties to the nanoparticles, PEGylation is a common approach to decrease immunogenicity and safeguard against the mononuclear phagocyte system [20]. Furthermore, to improve the delivery efficiency of nanoparticles, attention has been paid to conjugation with targeting ligands that bind to receptors overexpressed in the target tissues or cells.

Nanoparticles can be made of a variety of materials, namely polymers, lipids, and metals. Moreover, nano-antioxidants have been designed to overcome the oxidative degradation of organic and inorganic materials by slowing the overall rate of autoxidation [21]. Nevertheless, since the first clinical approval of Doxil^®^ in 1995, lipid-based nanoparticles remain the most prevalent class (33%) among nanomedicines on the market or under clinical trials [22]. In addition to AIDS-related Kaposi’s sarcoma (1995), this doxorubicin (DOX)-loaded PEGylated liposome (Doxil^®^) was FDA-approved in recurrent ovarian cancer (1998), metastatic breast cancer (2003), and multiple myeloma (2007) [23]; hence, it was a real breakthrough in cancer nanomedicine and lipid-based drug delivery systems.

Lipid-based nanoparticles include liposomes, solid lipid nanoparticles (SLNs), nanostructured lipid carriers (NLCs), and hybrid lipid–polymeric nanoparticles (Figure 1).

As drug delivery systems, lipid nanoparticles usually consist of hydrophilic or hydrophobic drug(s), lipids generally recognized as safe (GRAS), and surfactant(s) to form and stabilize the dispersion. Low toxicity potential, ease of preparation with limited use of organic solvents, and feasibility of scale-up production give them an edge over polymer and inorganic nanoparticles [24,25].

Regarding the development of actively targeted lipid nanoparticles, two general strategies exist: (i) one-pot assembly of lipids and targeting ligands and (ii) post-insertion of targeting ligands into preformed lipid nanoparticles. Still, the former is likely to cause the targeting ligands to be encapsulated or oriented to the nanoparticle core, making them inaccessible for receptor interaction [26]. Therefore, it comes as no surprise that post-insertion functionalization is preferred. In most cases, lipid nanoparticles are preformed by mixing structural lipids with PEG–lipids (e.g., 1,2-distearoyl-*sn*-glycero-3-phosphoethanolamine–polyethylene glycol, abbreviated to DSPE–PEG) containing different terminal groups, such as amino, carboxyl, maleimide, or NHS (e.g., DSPE–PEG–NH_2_, DSPE–PEG–COOH, DSPE–PEG–maleimide, DSPE–PEG–NHS, etc.), followed by modification with antibodies. Few examples of physical coating and streptavidin–biotin interaction can be found in antibody-functionalized drug-loaded lipid nanoparticles. While antibody fragments have been used occasionally, monoclonal antibodies (mAbs) are still at the forefront of antibody ligands. In 1975, hybridoma technology enabled the large-scale production of mAbs, which are antibodies against a single portion (epitope) of a specific antigen. To address clinical efficacy and safety issues, mAbs evolved from murine (suffix-omab) to chimeric (-ximab), humanized (-zumab), and, later, human (-umab). Whereas chimeric mAbs combine murine variable domains and human constant domains, in humanized mAbs, only CDRs are of non-human origin [27].

The utilization of antibody-conjugated lipid nanoparticles in anticancer drug delivery is discussed below, with several examples organized by nanoparticle type.

### 3.1. Antibody-Functionalized SLN

The lipids used to produce SLN are solid at both room temperature and human body temperature [28].

In 2011, the development of cationic SLN functionalized with a monoclonal antibody (mAb) directed against the epidermal growth factor receptor (EGFR) represented the first step in antibody-mediated strategies using drug-loaded SLN in cancer therapy. By using the microemulsion technique and EDC/NHS, Kuo and Liang fabricated anti-EGFR mAb-grafted cationic SLN encapsulating carmustine [29] and DOX [30] to target and inhibit the propagation of glioblastoma (GBM). When the concentration of cationic surfactants was 1 mM, the authors achieved the smallest particle size and maximal entrapment efficiency. Moreover, a high percentage of cacao butter in the lipid phase led to enhanced viability of human brain microvascular endothelial cells (HBMECs). Evidence from studies in EGFR-overexpressing U-87 MG glioma cells supported the antiproliferative effects of the targeted SLN, as well as the significance of surface anti-EGFR mAb for efficient drug delivery. Given that melanotransferrin (MTf) is found on HBMEC and U-87 MG cells, Kuo et al. [31] conjugated similar SLN with MTf antibody to deliver etoposide (ETP) across the blood–brain barrier for GBM therapy. Based on cell viability and Transwell assays, it is apparent that MTf antibody-conjugated ETP-loaded SLN had tolerable toxicity to HBMEC and human astrocytes, along with augmented transcytosis and growth inhibition of U-87 MG cells. As the design considerations become more complex, dual-targeted approaches for ETP delivery to GBM make the scene. More precisely, SLN contains two targeting ligands: MTf antibody plus tamoxifen to extend drug residence time within the brain parenchyma [32]; and anti-EGFR mAb and 83-14 mAb that bind strongly to the human insulin receptor on HBMEC prior to receptor-mediated endocytosis and subsequent brain delivery [33].

In another work, Kim et al. [34] performed studies in preclinical mouse models of lung or breast cancer, using paclitaxel (PTX)-containing low-density lipoprotein-mimicking SLN bearing the tumor-targeting antibodies cetuximab (CTX) or trastuzumab (TZM). The antitumor activity of these targeted SLN has exceeded that of the commercial formulations of PTX (Taxol^®^ and Genexol-PM), emphasizing the convenience of using nanocarriers and further adding targeting moieties to their surface. Together with the targeting ability, the antibody ligand may also have synergistic antitumor effects with the encapsulated drug. A case in point is the synergistic activity of the encapsulated oxaliplatin and tumor-necrosis-factor-related apoptosis-inducing ligand (TRAIL) mAb via SLN that elicited a 1.5-fold increase in cytotoxicity in HT-29 (colorectal cancer) cells compared to the free drug [35].

To improve internalization in breast cancer cells, Souto et al. [36] explored the streptavidin–biotin interaction to link cationic SLN with CAB51, a compact antibody against the human epidermal growth factor receptor 2 (HER2). The results showing that the targeting moiety enhanced cellular uptake in HER2-overexpressing BT-474 breast cancer cells inspired this author to adopt the same coupling strategy to engineer anti-HER2 CB11-modified cationic SLN for site-specific delivery of perillaldehyde 1,2-epoxide [37].

Triple-negative breast cancer (TNBC) is the most aggressive breast cancer subtype, accounting for approximately 10–15% of all cases. Attempts to cure TNBC are often frustrated by the lack of expression of estrogen, progesterone, and HER2 receptors. Some researchers sought to address this issue by functionalizing drug-loaded SLN with antibodies against receptors overexpressed in TNBC cells: exon v6-containing cluster of differentiation 44 isoform (CD44v6) [38], receptor for advanced glycation end-products (RAGE) [39], and death receptor 5 (DR5) [40]. Cavaco et al. [38] decorated PTX entrapped in SLN (SLN_PTX_) with polyethylene glycol-phosphatidylethanolamine and expected its hydroxyl groups to bind to the *N*-terminal amino groups of anti-CD44v6 mAb. Differences in MDA-MB-436 cell viability between the free drug and SLN_PTX_ can be attributed to the ability of nanoparticles to evade drug efflux transporters, thus promoting an increase in intracellular accumulation of PTX. However, the attached antibody failed to ameliorate the therapeutic efficacy of PTX compared to SLN_PTX_–PEG, thereby suggesting limitations in receptor-mediated endocytosis. By contrast, the internalization and cytotoxic effect of diallyl disulfide were significantly improved with an anti-RAGE antibody as the targeting ligand [39]. Additionally, an increase in cell death was observed following the treatment of MDA-MB-231 cells with anti-DR5 mAb-conjugated SLN of gamma-secretase inhibitor, which aligns with higher tumor regression in a breast cancer mouse model compared to non-targeted SLN or the free drug given intravenously (10 mg/kg) twice a week for four weeks [40]. Building upon the success of these in vitro and in vivo studies, Kumari et al. [41] hypothesized that introducing a second ligand, such as delta-like ligand 4 (DLL4) mAb, into this nanosystem would allow for the precise delivery to TNBC, reduction of side effects, and synergistic benefits.

A summary of these research papers is given in Table 1.

### 3.2. Antibody-Functionalized NLC

By including liquid lipids, NLC were endowed with great advantages over SLN, namely improved stability and high drug loading with minimal drug leakage during storage, arising from a less ordered crystalline structure [42].

The application of antibody-functionalized NLC in cancer chemotherapy has been somewhat scarce. Still, the recent literature contains some examples described in this section and listed in Table 2.

Taking advantage of vascular endothelial growth factor receptor 2 (VEGFR-2) overexpression in cancer cells and tumor neovasculature, Liu et al. [43] pioneered the “one-double targeting” strategy, meaning one ligand for double (tumor and vascular) targeting, with Flk-1(A-3) mAb. Due to the increased accumulation of docetaxel (DTX) in both tumor tissue and tumor vasculature, Flk-1 mAb-targeted DTX-loaded NLC showed better antitumor efficacy than non-targeted NLC and Duopafei^®^ (free DTX) against three human (HepG2, A549, and B16) cell lines and one malignant melanoma mouse model (dosage of 20 mg/kg).

For the targeting ligand coupled to DOX-loaded NLC by a post-insertion technique, Abdolahpour et al. [44] prepared a mAb directed against EGFRvIII. While evaluating the potential of the targeted NLC to increase cellular uptake, the authors observed that the uptake percentage in HC2 20d2/c (EGFRvIII-transfected NIH-3T3) cells was higher than that of NIH-3T3 cells, indicating that this antibody can specifically target EGFRvIII-overexpressing cells.

After developing Herceptin^®^ (TZM)-conjugated NLC containing DTX for HER2-positive breast cancer [45], Varshosaz et al. [46] employed rituximab to target CD20 receptors in lymphoma cells and selected the optimal NLC formulation co-loaded with curcumin and imatinib (a tyrosine kinase inhibitor). The optimal formulation containing lecithin and 25% of oleic acid was physically coated with a 20% rituximab solution, yielding an antibody coupling efficiency of 89 ± 0.15%. The treatment of Ramos (CD20-positive) B cells with the mixture of curcumin (15 µg/mL)/imatinib (5 µg/mL) (IC_50_ of 2.3 ± 0.1 µg/mL) and uncoated curcumin/imatinib-loaded NLC (IC_50_ of 2.9 ± 0.2 µg/mL) induced lower cytotoxicity than rituximab-coated curcumin/imatinib-loaded NLC with an IC_50_ of 1.4 µg/mL.

With the aim of attenuating off-target toxicity and overcoming resistance to monotherapy, NLC carrier systems and drug combination have garnered some interest in recent years. For instance, Guo et al. [47] highlighted synergistic combination therapy in lung cancer by preparing CTX-functionalized NLC co-encapsulating PTX and 5-demethylnobiletin. The presence of CTX resulted in the highest drug accumulation in the tumor tissue and the most remarkable tumor-growth inhibition from 1010.23 to 211.18 mm^3^ at the end of the study in PTX-resistant-lung-cancer-bearing mice. Upon intravenous (i.v.) injection every three days into mice bearing colorectal cancer xenografts, Liu et al. [48] verified a reduction of tumor growth without systemic toxicity through the decoration of NLC co-delivering irinotecan prodrug (2 mg/kg) and quercetin (2 mg/kg) with conatumumab/AMG 655 (anti-DR5 mAb).

Very recently, DTX-loaded NLC containing DSPE–PEG_2000_–maleimide was successfully coupled to thiolated bevacizumab (BVZ). The fabricated nanoformulation (BVZ-NLC-DTX) selectively induced cell death by apoptosis in vascular endothelial growth factor (VEGF)-overexpressing GBM cells (U-87 MG and A172), but not in peripheral blood mononuclear cells. Unlike free DTX (5 mg/kg, i.v.), BVZ-NLC-DTX (5 mg/kg, i.v.) was able to reduce up to 70% of tumor volume after a 15-day treatment in an orthotopic C6 glioma rat model [49].

### 3.3. Antibody-Functionalized Liposomes

Liposomes are a subtype of lipid-based nanoparticles mainly composed of phospholipids, which are amphiphilic molecules containing a hydrophilic head and a hydrophobic tail [50]. Although SLN and NLC were introduced as an alternative due to stability problems, liposomes are still the carrier of choice for many researchers with the goal of treating cancer with antibody-conjugated drug-loaded nanoparticles.

Targeted cancer therapies using antibody-functionalized liposomes are divided into angiogenesis-associated targeting, uncontrolled cell-proliferation targeting, and tumor-cell targeting that can be seen in subsequent sections.

#### 3.3.1. Angiogenesis-Associated Targeting

Angiogenesis is one of the hallmarks of cancer, as it allows the tumor to receive enough oxygen and nutrients to thrive [51]. In this strategy, liposomes encapsulating anticancer drugs are conjugated to antibodies that bind to receptors overexpressed in angiogenic endothelial cells, thus capitalizing on both antiangiogenic and cytotoxic effects to improve therapeutic efficacy. The suppression of blood-vessel growth within the tumor and normalization of tumor vasculature are the key mechanisms that underpin the antiangiogenic contribution to cancer cell killing [52]. To date, the angiogenic targets of antibody-functionalized liposomes for cancer therapy have been VEGF and its receptors, as well as matrix metalloproteinases.

Considering the prevalence of VEGF and VEGFR-2 in tumor cells and endothelium, as well as the low level of VEGF-induced endocytosis of VEGFR-2 in normal cells compared to tumor cells, it is possible to achieve efficient targeted delivery of nanoparticles to VEGF- and VEGFR-2-positive tumors. One prominent approach to angiogenesis-based targeting via VEGF/VEGFR-2 system involves targeting VEGF to inhibit its binding to VEGFR-2. Kuesters and Campbell [53] described the modification of PEGylated cationic liposomes with BVZ (anti-VEGF mAb) through the addition of neutravidin and biotinylated BVZ. From the experiments with human pancreatic cancer (Capan-1, HPAF-II, and PANC-1) and endothelial (MS1-VEGF and HMEC-1) cell lines, it became evident that BVZ increases or maintains the uptake of liposomes in the presence of VEGF but hinders non-specific cellular uptake when VEGF is absent. Recently, anti-VEGF mAb-conjugated PEGylated pH-sensitive liposomes were developed to augment the therapeutic efficacy of DTX in breast cancer while minimizing its side effects [54]. This nanosystem was a vast improvement over the marketed formulation Taxotere^®^, which showed a higher percentage of tumor burden of ~75% (vs. ~35%) in breast-tumor-bearing rats receiving a single dose equivalent to 2 mg/kg of free DTX. Additionally, Shein et al. [55] offered the first insight into the active targeting of liposomes to the brain tumor, using a mAb directed against VEGF. These targeted liposomes were then worked up into a more sophisticated liposomal formulation of cisplatin by coupling thiolated mAbs directed against VEGF and VEGFR-2 via maleimide chemistry [56]. Summarizing the in vitro data, the targeting moieties facilitated the uptake of the targeted liposomes in C6 and U-87 MG glioma cells, known for high VEGF and VEGFR-2 expression, leading to higher cytotoxicity than non-targeted and non-specific IgG–liposomes.

Only one of the articles reviewed [57] reported the use of an antibody targeting the membrane type 1-matrix metalloproteinase (MT1-MMP), particularly a Fab’ fragment derived from an anti-human MT1-MMP mAb (222-1D8). Compared to unmodified liposomes, modification with Fab’222-1D8 did not alter the tumor accumulation of PEGylated liposomes but appeared to accelerate their uptake in MT1-MMP-positive HT1080 fibrosarcoma cells after reaching the tumor via the EPR effect. The in vivo results also proved the utility of DOX-loaded Fab’222 1D8-modified liposomes from the point of view of antitumor activity.

#### 3.3.2. Uncontrolled Cell Proliferation Targeting

Here, drug-loaded liposomes are functionalized with antibodies directed against receptors involved in cancer-cell proliferation that are overexpressed in tumor cells, such as the human epidermal growth factor receptors and transferrin receptors. This strategy holds a particular promise for eradicating metastatic cells or small tumors that are devoid of new blood vessels.

To control the risk of idiosyncratic drug reactions and adverse reactions to afatinib combined with CTX, Lu et al. [58] developed liposomes loaded with that drug and coupled with CTX (IMC-C225 or Erbitux), a chimeric IgG1 mAb that binds to the extracellular domain of EGFR. In a non-small cell lung cancer (NSCLC) xenograft model with EGFR overexpression, those immunoliposomes showed a strongly enhanced ability for drug delivery and tumor growth inhibition.

Targetability to EGFR-overexpressing tumors can also be applied to stimuli-responsive nanocarriers. This is evidenced by sterically stabilized CTX-modified pH-responsive liposomes encapsulating gemcitabine (GEM) that were tested in A549 lung adenocarcinoma cells and NSCLC-bearing nude mice [59]. The apoptotic index in mice receiving the targeted liposomal formulation of GEM (160 mg/kg) was higher than that of other treatment groups and PBS-treated mice, and a good correlation between apoptosis and antitumor activity was found.

The absence of estrogen, progesterone and HER2 receptors has led to a concept of treating TNBC with EGFR-targeted liposomes carrying celecoxib [60], simvastatin [61], and PTX and piperine [62] with promising in vitro outcomes.

Following the functionalization of 5-fluorouracil (5-FU)-loaded liposomes with CTX, Petrilli et al. [63] investigated the influence of the administration route (topical or subcutaneous administration) on liposomal drug delivery in squamous cell carcinoma induced in immunosuppressed mice. According to the histological analysis, topical administration of 5-FU-loaded immunoliposomes using iontophoresis was more effective than subcutaneous injection in reducing cell proliferation and the remaining cells were less aggressive.

In another work [64], liposomes containing oxaliplatin were prepared and coupled via thioether linkage to whole CTX or Fab’ fragments derived from CTX. The site-directed conjugation to monovalent CTX-Fab’ fragments rendered liposomes with enhanced tumor accumulation (2916.0 ± 507.84 ng/g) compared to CTX (1546.02 ± 362.41 ng/g) or in the absence of targeting ligands (891.06 ± 155.1 ng/g), which also improved efficacy in mice inoculated with EGFR-positive colorectal cancer cells and treated with three i.v. 2.5 mg/kg doses at days 12, 15, and 18 post-cancer-cell implantation.

Eloy and colleagues [65] were the first to report on CTX-modified liposomes for the selective delivery of DTX to prostate cancer after preparing TZM-functionalized liposomes encapsulating rapamycin alone [66] or with PTX [67] to treat HER2-positive breast cancer. While TZM (anti-HER2 mAb) and rapamycin acted synergistically in SK-BR-3 cells, particularly via liposomes, the role of the antibody in mediating synergism between rapamycin and PTX was confirmed and attributed to improved cell uptake.

Also known as ErbB2 or CD340, HER2 regulates cell growth and proliferation and is one of the most used surface receptors to target liposomes for breast cancer and others, such as prostate cancer [68]. A growing number of therapeutic agents, namely bleomycin [69], DTX [70,71], curcumin and resveratrol [72], idarubicin [73], and epirubicin [74], have been proposed as payloads in breast cancer targeting, using TZM-conjugated liposomes. With liposomes linked to listeriolysin O and TZM, the amount of bleomycin needed to reduce tumor cell growth and viability is nearly 57,000-fold lower than the concentration required if the drug is administered extracellularly [69]. Based on the pharmacokinetic profile of TZM-coated vitamin E liposomes in male Sprague-Dawley rats, the half-life of DTX was 1.9 and 10 times longer than PEG-coated DTX-loaded vitamin E liposomes and a marketed formulation of DTX, respectively. Moreover, the area under the curve (97,740 ng.h/mL) was 3.47 times higher than that of DTX upon i.v. injection at an equivalent dose of 7 mg/kg [70]. Data from the in vivo distribution studies by Rodallec et al. [71] suggested that grafting TZM onto liposomes increases internalization rather than tumor localization. When compared to reference treatments (i.e., DTX + TZM or TZM emtansine), TZM-modified stealth liposomal DTX has greater efficacy in different breast cancer models (2D, 3D spheroids, and orthotopic xenograft mice). By coating liposomes with TZM, there was a dramatic increase in the antiproliferative effects of curcumin and resveratrol in HER2-overexpressing breast cancer cells [72]. The works by Amin et al. [73] and Khaleseh et al. [74] showed that such a drug delivery system could be a good choice for anthracyclines as well and provided a promising background for further in vivo studies.

The transferrin receptor (TfR) is another target for tumor-targeted delivery of liposomes using antibodies, of which OX26 mAb is the most representative ligand. By making use of the non-covalent (streptavidin–biotin) interaction, Schnyder et al. [75] described a new method of coupling biotinylated PEG–liposomes with streptavidin-linked OX26 mAb. The potential of biotinylated immunoliposomes to bypass P-glycoprotein (P-gp) in multidrug-resistant RBE4 brain capillary endothelial cells was manifested by a 2- to 3-fold enhancement in intracellular accumulation of daunomycin compared to the free drug. More recently, cisplatin-loaded PEGylated liposomes functionalized with OX26 mAb were found to be internalized in C6 cells more efficiently than non-functionalized liposomes, as well as having higher potency for enhanced therapeutic efficacy and less toxicity in brain-tumor-bearing Wistar rats [76]. In addition, Kim et al. [77] decorated cationic liposomes encapsulating temozolomide (TMZ) with an anti-TfR scFv to cross the blood–brain barrier and target GBM once in the brain parenchyma. The authors demonstrated that systemic administration of this novel formulation prolonged survival in mice bearing intracranial GBM. Furthermore, its improved efficacy in both TMZ-sensitive and TMZ-resistant tumors compared to standard TMZ was accompanied by a reduction in toxicity.

#### 3.3.3. Tumor-Cell Targeting

Tumor-cell targeting involves many of the aforementioned targets, as well as others specific to the type of cancer.

Due to the lack of oxygen, carbonic anhydrase IX (CA IX) is overexpressed in 80% of NSCLC. Yang’s group conjugated anti-CA IX antibody to DSPE–PEG–maleimide in the preformed DTX-loaded liposomes via sulfhydryl-reactive crosslinker chemistry [78]. A fluorescence-based flow cytometry assay was carried out in A549 cells and revealed that the binding affinity of targeted liposomes was 1.65-fold higher than non-targeted liposomes in CA IX-positive NSCLC cells. Later, by restraining tumor growth and prolonging the median survival time of up to 90 days in an orthotopic mouse model, they revalidated this strategy to deliver triptolide to human NSCLC via the pulmonary route [79].

The consistent expression of disialoganglioside (GD2) antigen in neuroblastoma cells and its limited expression in normal tissues outside the central nervous system opened the possibility of using this target in the treatment of neuroblastoma. To illustrate, liposomal formulations decorated with an anti-GD2 antibody and loaded with etoposide [80] and sepantronium bromide (YM155) [81] have been produced.

Whilst initial efforts to eradicate melanoma lesions focused on eliminating CD20-positive melanoma cancer-initiating cells with CD20 antibody-conjugated vincristine-loaded liposomes [82], recent attention was directed towards the expression of melanoma antigen A1 [83] and the programmed death-ligand 1 (PD-L1) [84] in melanoma cells. Since PD-L1 expression has been detected in 50% of gastric cancer patients, this receptor is also a target candidate for nanoparticle internalization in gastric cancer cells. Accordingly, the overall goal of improving the co-delivery of PTX and tariquidar (P-gp inhibitor) to multidrug-resistant SGC7901/ADR xenograft tumors could be achieved with anti-PD-L1 mAb-conjugated liposomes [85].

Within the scope of combination therapy, synergistic effects have been observed in pancreatic cancer cells when GEM was co-encapsulated with PTX in liposomes functionalized with the antibody fragment [86].

To refine the therapeutic efficacy of Doxil^®^ in hepatocellular carcinoma (HCC), Wang et al. [87] chose a post-insertion strategy to modify PEGylated liposomal DOX with the bivalent fragment HAb18 F(ab’)2 named Metuximab, which resulted in increased antitumor efficacy in CD147-overexpressing liver cancer cells and Huh-7 tumor xenografts. Concurrently, Lu et al. [88] constructed anti-CD44 antibody-modified liposomes to enhance the therapeutic index of timosaponin AIII in HCC. CD44 has also been recognized as a marker for cancer stem cells and is frequently expressed in several malignancies, including ovarian and colon cancers. As such, an anti-human CD44 mAb was employed to direct glycosylated PTX-loaded liposomes to CD44-positive ovarian cancer cells with great success [89].

Regarding colon cancer treatment, researchers proved the feasibility of coupling Doxil^®^ with anti-CD44 mAb [90] and anti-CD133 mAb [91] and accomplished higher therapeutic efficacy compared to other treatments (non-targeted Doxil^®^ and free DOX) against CD44-positive C-26 cells (in vitro and in vivo) and CD133-positive HT-29 cells, respectively. Based on preliminary findings [92], the progression of metastatic colorectal cancer could be haltered by 5-FU-loaded liposomes covalently linked to an antibody against the Frizzled 10 protein.

At the end of this subsection, Table 3 summarizes antibody-functionalized liposomes reported in papers from 2017 to 2022.

#### 3.3.4. Other Antibody-Functionalized Lipid-Based Nanoparticles

Another notable subset of lipid-based nanoparticles is commonly referred to as lipid–polymer hybrid (LPH) nanoparticles. These core–shell-type systems encompass attributes of lipids and polymers, as they consist of an inner lipid layer surrounding the polymer core that encapsulates the drug and an outer lipid–PEG layer, which prevents immune recognition and extends in vivo circulation time [93]. Since the outermost layer can be modified with different targeting ligands, such as antibodies, targeted cancer therapies using LPH nanoparticles (Table 4) are becoming increasingly popular.

The core of LPH nanoparticles usually comprises biodegradable polymers, such as polylactic-*co*-glycolic acid (PLGA). Hu et al. [94] conjugated LPH nanoparticles to a half-antibody against the carcinoembryonic antigen, overexpressed in 90% of pancreatic tumors. These authors confirmed the integrity of the PLGA core–lipid shell structure after internalization in BxPC-3 pancreatic cells, as well as the superior cytotoxicity of targeted PTX-loaded nanoparticles compared to non-targeted counterparts. In a therapeutic approach for HCC chemotherapy, a PLGA core containing adriamycin was covered by soybean lecithin/DSPE–PEG in the shell and further coated with anti-EGFR Fab’ [95]. More recently, Wei et al. [96] designed salinomycin-loaded PLGA-lipid nanoparticles linked to anti-CD44 Fab’ to eliminate prostate-cancer-initiating cells.

To achieve the targeted co-delivery of cisplatin and 5-FU to HER2-overexpressing esophageal adenocarcinoma, TZM was conjugated to the surface of LPH nanoparticles formulated by the unique combination of DSPE–PEG–COOH, soy phosphatidylcholine, and poly(*ε*-caprolactone) [97].

Compared to the whole antibody molecule, anti-CD33 Fab’-decorated LPH nanoparticles had longer circulation times in naïve mice and enabled higher levels of 1-*β*-D-arabinofuranosylcytosine in blood. Unfortunately, when it comes to prolonging the survival of leukemic mice, the incorporation of a pH-sensitive copolymer made of dioctadecyl, *N*-isopropyl acrylamide and methacrylic acid did not add any benefit to the formulation [98].

Differently, Leung et al. [99] reported the chemical conjugation of organic–inorganic hybrid nanovesicles to anti-EGFR mAbs, which retain the ability to bind to EGFR and inhibit A431 epidermoid carcinoma cell proliferation. It is noteworthy that the surface polysiloxane network makes these lipid nanovesicles more morphologically stable than conventional liposomes. Another alternative to liposomes are niosomes, that is to say, lipid nanovesicles based on non-ionic surfactants [100]. Anti-CD123 antibodies conjugated to maleimide–PEG_2000_–DSPE were incorporated into daunorubicin-loaded niosomes via a post-insertion technique for treating acute myeloid leukemia. The obtained niosomes were tested in NB4 and THP-1 cells and CD123-overexpressing leukemic mice, showing higher cytotoxicity and prolonged survival time [101].

Finally, Zhai and colleagues [102] proposed liquid-crystalline lipid carriers (i.e., cubosomes) as targeted delivery vehicles for PTX in aggressive ovarian cancer. By adding DSPE–PEG–maleimide to the formulation, it was possible to functionalize cubic-phase lipid nanoparticles with EGFR 528 Fab’. Still, this surface modification did not improve their performance either in vitro or in vivo compared to PTX-loaded cubosomes without the antibody.

### 3.4. Clinical Trials

Although the literature portrays a picture of potential therapeutic benefits and low systemic toxicity, the number of antibody-functionalized nanoparticles available to cancer patients is drastically below expectation, partially owing to a translational gap between preclinical and clinical studies. None of these nanoconjugates has been approved so far, but several candidates for the treatment of solid tumors have entered the clinical testing phase, including (i) anti-TfR scFv-modified liposomes carrying plasmid DNA (e.g., SGT-53 and SGT-94) [103,104]; (ii) a dendritic cell-targeted liposomal vaccine called Lipovaxin-MM [105]; (iii) bacterially derived mini- or nano-cells (EnGeneIC delivery vehicles or EDV) coupled to bispecific antibodies (e.g., Erbitux^®^EDVs_PAC_, EGFR(V)-EDV-Dox, and EEDVsMit) [106,107]; and (iv) antibody-directed liposomes encapsulating chemotherapeutic drugs (e.g., MM-310, MM-302, C225-ILs-Dox, and MCC-465) [108,109,110,111].

Whereas MM-310 is an anti-ephrin A2 scFv attached to liposome encapsulating a DTX prodrug, MM-302 and C225-ILs-dox stem from the modification of a liposomal formulation resembling Doxil^®^ with post-inserted anti-HER scFv and CTX Fab’, respectively.

A study [112] using multiple tumor-xenografted mice was the basis for initiating a phase 1 clinical trial (NCT03076372) to assess the safety and efficacy of MM-310 in multiple solid tumors. Kamoun et al. [113] determined its preclinical efficacy in four ephrin A2-positive patient-derived xenograft models of bladder cancer, either as a monotherapy or in combination with GEM.

A phase 1 dose-escalation study (NCT01304797) evaluated the safety, tolerability, and pharmacokinetics of MM-302 as a monotherapy, in combination with TZM, or TZM plus cyclophosphamide in patients with advanced HER2-positive breast cancer. This study is now completed, and the recommended dose for a phase 2 study was MM-302 30 mg/m^2^ in combination with 6 mg/kg TZM every 3 weeks. In addition, Espelin et al. [114] demonstrated the synergistic antitumor activity of MM-302 combined with TZM in human xenograft models of breast and gastric cancer. This work provided the preclinical foundation for the HERMIONE clinical trial (NCT02213744) in anthracycline-naïve patients with locally advanced/metastatic HER2-positive breast cancer patients receiving MM-302 plus TZM versus the chemotherapy of physician’s choice (GEM, capecitabine, or vinorelbine) plus TZM. Unfortunately, in 2016, Merrimack Pharmaceuticals decided to halt further development of MM-302 due to negative outcomes. Later, in 2019, the observation of cumulative peripheral neuropathy in a phase 1 study precluded the possibility of continuing the development of MM-310.

After a phase 1 dose-escalation trial (NCT01702129) in advanced solid tumors, C225-ILs-Dox was clinically evaluated in patients with relapsed or refractory high-grade gliomas (NCT03603379). Although the delivery of C225-ILs-dox to glioblastoma tissue was demonstrated, no other definitive conclusions can be drawn from this trial, as there was no control group, and only a few patients were treated [115]. The DOX-loaded PEGylated immunoliposome termed MCC-465 utilizes the F(ab’)_2_ fragment of human GAH mAb as the targeting ligand, which positively reacts to over 90% of gastric cancer tissues but negatively to all normal tissues. A phase 1 study of MCC-465 conducted in patients with metastatic gastric cancer established the recommended dose of 32.5 mg/m^2^ for a phase 2 study with a 3-week schedule, but recent updates are missing [111].

## 4. Conclusions

As drug delivery systems, lipid-based nanoparticles offer many advantages, including biocompatibility, self-assembly, enhancement of drug solubility and bioavailability, high loading capacity, ease of production and modulation of drug release, cost-effectiveness, and feasibility of scale-up. Consequently, this type of nanocarrier has gained significant attention from researchers in both academia and industry over the past two to three decades. With the advances in bioconjugation and antibody engineering techniques, the development of antibody-functionalized lipid nanoparticles for anticancer drug delivery has been widespread, generating promising results in vitro and in small animal models. However, only a small number of these targeted nanomedicines have progressed into early clinical evaluation, and none has reached the market yet. Therefore, understanding the reasons for their underperformance in clinical settings is critical to guide future directions.

First, the physical characterization of nanoparticles deserves a robust strategy and suitable protocols, as it could help accelerate the shift from lab to industrial-scale production and anticipate their behavior in vivo [116]. In the quest for new knowledge regarding nanoparticle interactions with biological components, the use of different tools, such as computational analysis, mathematical modeling, and microfluidic platforms, should also be intensified.

Still, we must be aware that only a tiny fraction (less than 14 out of 1 million) of nanoparticles with an active targeting moiety can enter the solid tumor after intravenous injection [117], so it is convenient to consider local administration (e.g., intratumoral injection) whenever possible [118].

Many recent studies have simply outlined the toxic effects of nanoparticles, but only a few have systematically addressed their potentially adverse impact on mammalian target organs and after chronic exposure [119,120]. In addition to the assessment of bulk material safety, researchers should perform a thorough characterization of the nanoparticles, including analysis of batch-to-batch differences regarding consistency, stability, sterility and endotoxin quantification, blood contact properties, and in vivo cytotoxicity and immunotoxicology, to prevent toxicity-related clinical failure [121]. Accordingly, standardized guidelines to obtain nanotoxicological profiles should be implemented and followed to evaluate the potential risk in patients [122].

The discrepancy between preclinical and clinical observations is also explained by artificial rodent tumors that fail to mimic human tumors regarding transport and delivery. Accordingly, the development of in vivo models that closely reflect human cancer biology (e.g., high-fidelity patient-derived xenografts, humanized or genetically engineered mouse models, etc.) will certainly increase their predictive power.

Besides pathophysiological differences between animal model species and humans, heterogeneity amongst patients and tumor types or within the same patient over time, in terms of molecular and phenotypic features and the extent of the EPR effect, can also limit the clinical success of nanomedicines. Therefore, it is recommended to quantify the EPR effect in patient tumors to identify those who would benefit most from the treatment with therapeutic nanoparticles. Selecting the right patients is pivotal to the success of clinical trials, as well as finding the most suitable payload/combination regimen and trial size.

Despite the lack of translation progress, active targeting of nanoparticles remains an exciting research topic for cancer therapy, as antibody-conjugated lipid nanoparticles incorporating novel classes of therapeutics (e.g., RNA-based and gene editing therapeutics) other than cytotoxic drugs and kinase inhibitors are emerging. Looking to the future, one can expect more interdisciplinary research collaborations to open new avenues for the development of tumor-targeted lipid nanoparticle systems and, thus, hasten their clinical validation.

## Figures and Tables

**Figure 1 pharmaceutics-15-00216-f001:**
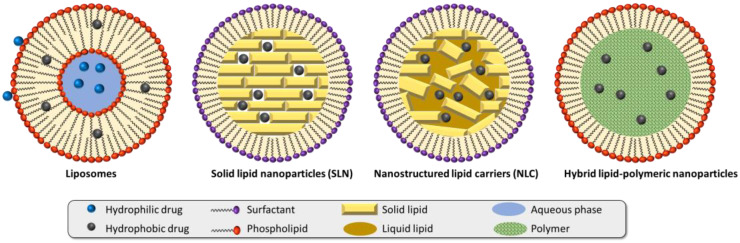
Different types of lipid nanoparticles. Adapted with permission from [24]. Copyright 2022, John Wiley & Sons.

**Table 1 pharmaceutics-15-00216-t001:** Antibody-functionalized SLN for anticancer drug delivery.

Lipids	Drug	Ligands	Coupling Method	Cancer Cell (In Vitro)	Cancer (In Vivo)	Ref.
Cacao butter, SA DSPE–PEG_2000_–COOH	Carmustine	EGFR mAb	Carbodiimide	U-87 MG (glioblastoma)	.	[29]
Cacao butter, SA DSPE–PEG_2000_–COOH	Doxorubicin	EGFR mAb	Carbodiimide	U-87 MG (glioblastoma)	-	[30]
Tripalmitin, cacao butter Cardiolipin, DSPE–PEG_2000_–COOH	Etoposide	MTf Ab	Carbodiimide	U-87 MG (glioblastoma)	-	[31]
Compritol^®^ 888 ATO, tripalmitin, Ch, SA, DSPE–PEG_2000_–COOH	Etoposide	MTf Ab Tamoxifen	Carbodiimide	U-87 MG (glioblastoma)	-	[32]
Compritol^®^ 888 ATO, tripalmitin Stearic acid, DSPE–PEG_2000_–COOH	Etoposide	EGFR mAb 83-14 mAb	Carbodiimide	U-87 MG (glioblastoma)	-	[33]
Cholesteryl oleate, triolein Ch, DOPE, DC-cholesterol	Paclitaxel	Cetuximab Trastuzumab	Maleimide	NCI-H1975, NCI-H1650 NCI-H520, PC9, SK-BR-3	Lung Breast	[34]
SA	Oxaliplatin	TRAIL mAb	Carbodiimide	HT-29 (colorectal)	-	[35]
Compritol^®^ 888 ATO	-	CAB51	Streptavidin-biotin	MCF-7, BT-474 (breast)	-	[36]
Compritol^®^ 888 ATO	Perillaldehyde 1,2-epoxide	CB11	Streptavidin-biotin	MCF-7 (breast)	-	[37]
Precirol^®^ ATO 5, PEG–PE	Paclitaxel	CD44v6 mAb	Hydroxyl-amino	MDA-MB-436 (TNBC)	-	[38]
Palmitic acid	Diallyl disulfide	RAGE Ab	Carbodiimide	MDA-MB-231 (TNBC)	-	[39]
SA	DAPT	DR5 mAb	Carbodiimide	MDA-MB-231	Breast	[40]

CD44v6, CD44 variant 6; Ch, cholesterol; DAPT, N-[N-(3,5-difluorophenacetyl)-L-alanyl]-*S*-phenylglycine *t*-butyl ester; DC-cholesterol, 3β-[N-(N′,N′-dimethylaminutesoethane)-carbamoyl] cholesterol hydrochloride; DOPE, 1,2-dioleoyl-*sn*-glycero-3-phosphoethanolamine; DR5, death receptor 5; DSPE, 1,2-distearoyl-*sn*-glycero-3-phosphoethanolamine; EGFR, epidermal growth factor receptor; HER2, human epidermal growth factor receptor 2; MTf, melanotransferrin; PEG, polyethylene glycol; PEG–PE, polyethylene glycol–phosphatidylethanolamine; RAGE, receptor for advanced glycation end-products; SA, stearic acid; TNBC, triple-negative breast cancer; TRAIL, tumor-necrosis-factor-related apoptosis-inducing ligand.

**Table 2 pharmaceutics-15-00216-t002:** Antibody-functionalized NLC for anticancer drug delivery.

Lipids	Drug	Ligands	Coupling Method	Cancer Cell (In Vitro)	Cancer (In Vivo)	Ref.
SA Glyceryl monostearate Middle chain triglycerides DSPE–PEG–NH_2_	Docetaxel	Flk-1(A-3) mAb	BS3 crosslinker	HepG2 (HCC) A549 (lung) B16 (melanoma)	Melanoma	[43]
SA Oleic acid DSPE–PEG_2000_–NHS	Doxorubicin	EGFRvIII mAb	Amine-reactive crosslinker	HC2 20d2/c	-	[44]
Cholesterol Castor oil SA Fatty amines NHS–PEG_3K_–maleimide	Docetaxel	Trastuzumab	Maleimide	MDA-MB-468 BT-474 (breast)	-	[45]
Glyceryl monostearate or lecithin Oleic acid or Labrafac^®^	Curcumin Imatinib	Rituximab	Ionic adsorption	Jurkat and Ramos (lymphoma)	-	[46]
Oleic acid Compritol^®^ 888 ATO Soybean phosphatidylcholine DSPE–PEG–maleimide	Paclitaxel 5-Demethylnobiletin	Cetuximab	Maleimide	A549	Lung	[47]
Glycerin monostearate Soybean oil	Irinotecan prodrug Quercetin	Conatumumab	Carbodiimide	HT-29	Colorectal	[48]
DSPE–PEG_2000_–maleimide Caprylic/capric triglyceride PEG–40 hydrogenated castor oil	Docetaxel	Bevacizumab	Maleimide	U-87 MG A172	Glioblastoma	[49]

BS3, bis(sulfosuccinimidyl)suberate; DSPE, 1,2-distearoyl-*sn*-glycero-3-phosphoethanolamine; EGFR, epidermal growth factor receptor; HCC, hepatocellular carcinoma; mAb, monoclonal antibody; NHS, *N*-hydroxysuccinimide; PEG, polyethylene glycol; SA, stearic acid.

**Table 3 pharmaceutics-15-00216-t003:** Antibody-functionalized liposomes for anticancer drug delivery.

Lipids	Drug	Ligands	Coupling Method	Cancer Cell (In Vitro)	Cancer (In Vivo)	Ref.
Soya lecithin, Ch, DOPE CHEMS, DSPE–PEG–COOH	Docetaxel	VEGF mAb	Carbodiimide	MCF-7	Breast	[54]
HSPC, Ch, DSPE–PEG_2000_ DSPE–PEG_2000_–maleimide	Afatinib	Cetuximab	Maleimide	A549 H1975	NSCLC	[58]
HSPC, DSPC, Ch, DSPE–PEG DSPE–PEG–maleimide	Simvastatin	Cetuximab	Maleimide	MDA-MB-231	TNBC	[61]
HSPC, Ch (TPGS and TPGS–COOH)	Paclitaxel Piperine	Cetuximab	Carbodiimide	MDA-MB-231 (TNBC)	-	[62]
DSPC, Ch DSPE–PEG–maleimide	5-Fluorouracil	Cetuximab	Maleimide	A431 B16F10	Squamous cell carcinoma	[63]
SPC, Ch, DSPE–PEG_2000_ DSPE–PEG–maleimide	Docetaxel	Cetuximab	Maleimide	DU145 PC-3 (prostate)	-	[65]
SPC, Ch, DSPE–PEG_2000_ DSPE–PEG–maleimide	Rapamycin	Trastuzumab	Maleimide	MDA-MB-231 SK-BR-3 (breast)	-	[66]
SPC, Ch, DSPE–PEG_2000_ DSPE–PEG–maleimide	Rapamycin Paclitaxel	Trastuzumab	Maleimide	4T1 SK-BR-3	Breast	[67]
SPC, Ch DSPE–PEG_2000_–NHS	Doxorubicin Simvastatin	Trastuzumab	Amine-reactive crosslinker	PC3	Prostate	[68]
Phosphatidylcholine Phosphatidylglycerol Ch, maleimide–PEG	Docetaxel	Trastuzumab	Maleimide	SK-BR-3	Breast	[71]
SPC, Ch, DSPE–PEG DSPE–PEG–maleimide	Idarubicin	Trastuzumab	Maleimide	MCF-7 SK-BR-3 (breast)	-	[73]
DOPE, Ch	Epirubicin	Trastuzumab	Carbodiimide	MCF-7, MDA-MB-453 BT-20 (breast)	-	[74]
Lecithin, Ch, DSPE–PEG_2000_ DSPE–PEG_2000_–maleimide	Cisplatin	OX26 mAb	Maleimide	C6	Glioma	[76]
SPC, DSPE–PEG_2000_ DSPE–PEG_2000_–maleimide	Triptolide	CA IX Ab	Maleimide	A549	NSCLC	[79]
DPPC, Ch, DSPE–PEG_2000_ DSPE–PEG_2000_–maleimide	Sepantronium bromide	GD2 Ab	Maleimide	IMR32 KCNR	Neuroblastoma	[81]
HSPC, Ch, DSPE–PEG_2000_ DSPE–PEG_2000_–maleimide	Doxorubicin	scFv G8 Hyb3	Maleimide	MZ2Mel43, G43 Mel2A, Mel78	Melanoma	[83]
HSPC, Ch, DSPE–PEG_2000_ DSPE–PEG_2000_–maleimide	Doxorubicin	PD-L1 mAb	Maleimide	B16-OVA	Melanoma	[84]
Egg phosphatidylcholine DOPE, DSPE–PEG_2000_ DSPE–PEG_2000_–maleimide	Paclitaxel Tariquidar	PD-L1 mAb	Maleimide	SGC7901/ADR	Gastric	[85]
DPPC, Ch, DSPE–PEG_2000_ DSPE–PEG_2000_–maleimide	Paclitaxel	Ab fragment	Maleimide	BxPC3 (pancreatic)	-	[86]
DSPE–PEG_2000_–maleimide	Doxil^®^	Metuximab	Maleimide	Huh-7, HepG2 HCC 3736	HCC	[87]
DSPC, DSPE–PEG_2000_ DSPE–PEG_2000_–maleimide	Timosaponin AIII	CD44 Ab	Maleimide	HepG2	HCC	[88]
DPPC, Ch, mPEG–DSPE DSPE–PEG_2000_–maleimide	Glycosylated paclitaxel	CD44 mAb	Maleimide	SK-OV-3, OVCAR-3 OVK18	Ovarian	[89]
mPEG_2000_–DSPE DSPE–PEG_3400_–NHS	Doxil^®^	CD133 mAb	Amine-reactive crosslinker	HT-29 (colorectal)	-	[91]
Ch, Phosphatidylcholine Stearylamine, DSPE–PEG_2000_ DSPE–PEG_2000_–COOH	5-Fluorouracil	FZD10 Ab	Carbodiimide	CaCo-2 CoLo-205 (colorectal)	-	[92]

CA, carbonic anhydrase; CD, cluster of differentiation; Ch, cholesterol; CHEMS, cholesteryl hemisuccinate; DOPE, 1,2-dioleoyl-*sn*-glycero-3-phosphoethanolamine; DPPC, 1,2-dipalmitoyl-*sn*-glycero-3-phosphocholine; DSPC, 1,2-distearoyl-*sn*-glycero-3-phosphocholine; DSPE, 1,2-distearoyl-*sn*-glycero-3-phosphoethanolamine; FZD10, frizzled 10 protein; GD2, disialoganglioside; HCC, hepatocellular carcinoma; HSPC, hydrogenated soy phosphatidylcholine; NHS, *N*-hydroxysuccinimide; NSCLC, non-small cell lung cancer; PD-L1, programmed death-ligand 1; PEG, polyethylene glycol; SPC, soy phosphatidylcholine; TNBC, triple-negative breast cancer; TPGS, PEGylated vitamin E succinate; VEGF, vascular endothelial growth factor.

**Table 4 pharmaceutics-15-00216-t004:** Other antibody-functionalized lipid nanoparticles for anticancer drug delivery.

Nanocarrier	Drug	Ligands	Coupling Method	Cancer Cell (In Vitro)	Cancer (In Vivo)	Ref.
Lipid–polymer hybrid NP	Paclitaxel	CEA half-Ab	Maleimide	BxPC-3 XPA-3 (pancreatic)	-	[94]
Lipid–polymer hybrid NP	Adriamycin	EGFR Fab’	Maleimide	SMMC-7721 HepG2 Huh7	HCC	[95]
Lipid–polymer hybrid NP	Salinomycin	CD44 Fab’	Maleimide	DU145 22RV1	Prostate	[96]
Lipid–polymer hybrid NP	Cisplatin 5-Fluorouracil	Trastuzumab	Carbodiimide	BE-3	Esophageal	[97]
Lipid–polymer hybrid NP	Ara-C	CD33 mAb or Fab’	Maleimide	HL-60	AML	[98]
Lipid nanovesicles	-	EGFR mAb	Maleimide	DU145 (prostate) A431 (epidermoid carcinoma)	-	[99]
Niosome	Daunorubicin	CD123	Maleimide	NB4 THP-1	AML	[101]
Cubosome	Paclitaxel	EGFR 528 Fab’	Maleimide	HEY	Ovarian	[102]

AML, acute myeloid leukemia; Ara-C, 1-*β*-D-arabinofuranosylcytosine; CD, cluster of differentiation; CEA, carcinoembryonic antigen; EGFR, epidermal growth factor receptor; HCC, hepatocellular carcinoma; NP, nanoparticle.

## Data Availability

Not applicable.

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
