# Peer review of "Lipid Nanoparticles Functionalized with Antibodies for Anticancer Drug Therapy"

_pharmaceutics, 2023, doi:10.3390/pharmaceutics15010216_

Round 1

Reviewer 1 Report

1.It is suggested to add new research studies about the toxicological concerns.
2.what is the suggestion of this study for future works?
3.Please discuss and compare your results with previous works and add suggestions including Nano antioxidants based
methods.
4.It will be better to add the role of mitochondria targeting and also Nfk-b pathway.
5.Please add details for time period and dose selection from literatures.
6.More references for the discussion part of manuscript and update and bold your study novelty should be added: e.g.,
-DOI: 10.1016/j.pestbp.2020.104586

-DOI: 10.3390/pharmaceutics13040549

Author Response

We appreciate the time and effort in providing feedback on our manuscript. We have carefully reviewed the comments and incorporated most of the suggestions made. The revised manuscript is an MS Word file showing tracked changes in red.

Our explanations are presented below in a point-by-point response. All line numbers refer to the revised manuscript file.

  • Comment 1: It is suggested to add new research studies about the toxicological concerns.

Response: We agree with this and have added the suggested content to the manuscript (lines 627-635).

  • Comment 2: What is the suggestion of this study for future works?

Response: This study suggests that the active targeting of nanoparticles will remain an exciting research topic for cancer therapy. We believe future works will focus on functionalizing different types of nanoparticles and incorporating novel classes of therapeutics (RNA-based and gene editing therapeutics) other than cytotoxic drugs and kinase inhibitors. Antibodies (full-length or fragments) will probably remain the most frequently used ligands to target tumor cells. Moreover, as explained in Section 5., interdisciplinary research teams can make a major contribution to clinical translation by deepening our understanding of nanoparticle physical characteristics and interaction with biological components; developing in vivo models that closely reflect human cancer biology; exploring other administration routes to address poor pharmacokinetics and insufficient drug accumulation in the tumor, etc.

  • Comment 3: Please discuss and compare your results with previous works and add suggestions including Nano antioxidants based methods.

Response: Thank you for your comment. We are still conducting our preliminary steps in developing blank NLC dispersions. The antibody-functionalized drug-loaded formulations will then be studied to evaluate their technological characteristics, stability, and in vitro and in vivo efficacy and toxicity. We expect to be reporting our first results in the next year and understanding whether they are consistent with previous findings or not. Finally, we briefly mentioned nanoantioxidants when introducing nanoparticles in Section 3 (lines 133 and 134).

  • Comment 4: It will be better to add the role of mitochondria targeting and also Nfk-b pathway.

Response: We appreciate the reviewer’s comment. We agree that targeting cancer cell mitochondria could potentially be an alternative approach to treating this disease. To date, nanoformulations have been made mitochondriotropic by conjugating them with lipophilic cations, peptides, or aptamers. Since the use of antibodies as mitochondria-targeting moieties is yet to be reported, we believe this subject does not come within the scope of this review. As regards NF-κB targeting, we already provided some examples of antibody-functionalized lipid nanoparticles encapsulating drugs that were proven to inhibit the NF-κB pathway (ref. 79, 88, 96, etc.).

  • Comment 5: Please add details for time period and dose selection from literatures.

Response: Thank you for pointing this out. We added more information on the experimental conditions (duration of treatment and dose selection) throughout the manuscript.

  • Comment 6: More references for the discussion part of manuscript and update and bold your study novelty should be added: e.g., DOI: 10.1016/j.pestbp.2020.104586, DOI: 10.3390/pharmaceutics13040549.

Response: We appreciate this comment. Although we acknowledge the potential of quercetin-loaded NLC to mitigate paraquat-induced toxicity in human lymphocytes, these nanoparticles have not been functionalized with antibodies or designed to target cancer cells. Accordingly, the first content (DOI: 10.1016/j.pestbp.2020.104586) is, unfortunately, beyond the scope of this review. Following the second suggestion (DOI: 10.3390/pharmaceutics13040549), we added a comment on the importance of the physical characterization of nanoparticles for scale-up production and clinical translation (lines 616-619).

Reviewer 2 Report

Overall, this manuscript provides a good overview of the use of lipid-based nanoparticles for anti-cancer drug delivery, particularly those functionalized with antibodies. However, a few suggestions for improvement could include:

1.     Clarifying the specific types of cancer that these nanoparticles have shown promise in treating.

2.     Providing more detail on the mechanisms by which the nanoparticles improve therapeutic efficacy, such as by targeting specific receptors overexpressed in cancer cells.

3.     Including more information on the challenges and limitations of using these nanoparticles in clinical practice, such as low translation success and potential systemic toxicity.

4.     Discussing potential future directions for the development and use of these nanoparticles, including any ongoing or planned clinical trials.

5.     Providing examples or case studies of successful applications of these nanoparticles in cancer treatment to illustrate their effectiveness.

Overall, these suggestions could help to provide a more comprehensive and nuanced understanding of the current state of anti-cancer drug delivery using lipid-based nanoparticles.

Author Response

We appreciate the time and effort in providing feedback on our manuscript. We have carefully reviewed the comments and incorporated most of the suggestions made. The revised manuscript is an MS Word file showing tracked changes in red.

Our explanations are presented below in a point-by-point response. All line numbers refer to the revised manuscript file.

Reviewer #2

  • Overall, this manuscript provides a good overview of the use of lipid-based nanoparticles for anti-cancer drug delivery, particularly those functionalized with antibodies. However, a few suggestions for improvement could include: (continued below)

Response: The reviewer’s positive feedback is greatly appreciated.

  • Comment 1: Clarifying the specific types of cancer that these nanoparticles have shown promise in treating.

Response: We agree that we should have clarified the types of cancer that these nanoparticles have shown promise in treating. Only a couple of actively targeted lipid nanoparticles have been developed to treat hematological malignancies, while the most common indication has been solid tumors (line 17).

  • Comment 2: Providing more detail on the mechanisms by which the nanoparticles improve therapeutic efficacy, such as by targeting specific receptors overexpressed in cancer cells.

Response: As suggested by the reviewer, the mechanisms by which nanoparticles actively target specific receptors overexpressed in cancer cells are now briefly detailed. The main differences in targeting internalizing and non-internalizing receptors are also provided (lines 46-50; 53-56).

  • Comment 3: Including more information on the challenges and limitations of using these nanoparticles in clinical practice, such as low translation success and potential systemic toxicity.

Response: Thank you for your suggestions. We have rewritten the conclusion to include other factors that could contribute to clinical success, namely safety concerns (lines 627-635) and clinical study design (lines 648-650).

  • Comment 4: Discussing potential future directions for the development and use of these nanoparticles, including any ongoing or planned clinical trials.

Response: Thank you for pointing this out. To the best of our knowledge, AR160 is the only antibody-functionalized drug-containing nanoparticle undergoing a cancer clinical trial (NCT03003546). AR160 is being tested for relapsed or refractory B-cell non-Hodgkin lymphoma, and the estimated study completion date is 2023. However, AR160 is a nanoparticle albumin-bound paclitaxel coated with rituximab and this review aims to provide readers with specific information on lipid-based nanocarriers. Updates on planned clinical trials related to this type of nanoparticle are currently missing.

  • Comment 5: Providing examples or case studies of successful applications of these nanoparticles in cancer treatment to illustrate their effectiveness.

Response: We agree with this and have added the suggested content to the manuscript (lines 567-582; 588-590).

  • Overall, these suggestions could help to provide a more comprehensive and nuanced understanding of the current state of anti-cancer drug delivery using lipid-based nanoparticles.

Response: Thank you. We took your suggestions as a valuable opportunity to improve the quality of our manuscript.

Additional information

In addition to the above changes, we have repositioned the tables after adding new text. Added references (ref. 4, 6, 21, 112-116, and 119-122) are highlighted in the reference list.
